# ROCM: RLHF on consistency models

## Abstract

Diffusion models have revolutionized generative modeling in continuous domains like image, audio, and video synthesis. However, their iterative sampling process leads to slow generation and inefficient training, challenges that are further exacerbated when incorporating Reinforcement Learning from Human Feedback (RLHF) due to sparse rewards and long time horizons. Consistency models address these issues by enabling single-step or efficient multi-step generation, significantly reducing computational costs. In this work, we propose a direct reward optimization framework for applying RLHF to consistency models, incorporating distributional regularization to enhance training stability and prevent reward hacking. We investigate various $f$-divergences as regularization strategies, striking a balance between reward maximization and model consistency. Unlike policy gradient methods, our approach leverages first-order gradients, making it more efficient and less sensitive to hyperparameter tuning. Empirical results show that our method achieves competitive or superior performance compared to policy gradient based RLHF methods, across various automatic metrics and human evaluation. Additionally, our analysis demonstrates the impact of different regularization techniques in improving model generalization and preventing overfitting.

## 1 Introduction

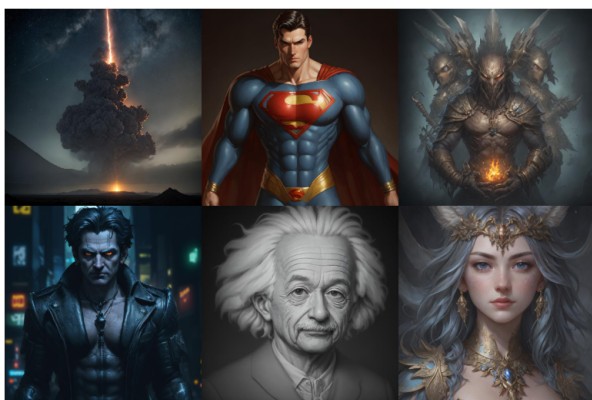

Figure 1: **Generation samples:** Sample Images generated by our method trained on HPSv2 as reward model.

Diffusion models have brought about significant advancements in the modeling of continuous domains, including chemical molecule design Zeni et al. (2024), audio generation Lemercier et al. (2024), text-to-image synthesis Rombach et al. (2022), and video generation Lin et al. (2024). These models have demonstrated remarkable success across various applications, showcasing their versatility and potential. However, a notable challenge with diffusion models is their slow generation process. The iterative nature of the diffusion process means that each sample generation involves multiple steps, often making it difficult to train these models in an end-to-end manner. Researchers usually have to resort to approximations in the learning pipeline Black et al. (2024); Wallace et al. (2023) or endure extensive training times to fine-tune diffusion models effectively Clark et al. (2024).

This challenge becomes even more pronounced when applying reinforcement learning (RL) pipelines to fine-tune diffusion models. In these scenarios, rewards are provided only at the final step of the generation process, and as the time horizon increases the resulting sparse reward signals can significantly hinder the training performance. To tackle this problem, we turn our attention to the performance of Reinforcement Learning from Human Feedback (RLHF) when applied

to consistency models Song et al. (2023). Consistency models, in contrast to diffusion models, offer the advantage of efficient generation with a small number of steps. In practice, they can produce competitive results within 4-8 steps, compared to the 20-50 steps typically required by diffusion models, thus addressing the issue of slow generation to a large extent. A key observation in our study is that simply maximizing the reward in the RLHF pipeline often leads to overfitting and reward hacking, as the trained model diverges significantly from the original model. Reward hacking Stiennon et al. (2022) arises when the generation policy strays too far from the reference model, producing samples that are substantially different from those used to train the reward model. In such an out-of-distribution regime, a high reward does not necessarily indicate high-quality outputs. Over-optimizing for the reward can therefore lead to poor-quality images that receive artificially inflated scores. To counter this, it is common to alter the objective to not only maximize the reward but also to minimize the divergence between the current model and the reference model distributions. The reference model is generally set to the base model at the beginning of training. By experimenting with different $f$-divergence measures, we find that this form of regularization helps stabilize the training process, preventing the model from degenerating into reward hacking and ensuring more robust performance across various metrics.

While prior research has applied methods such as Proximal Policy Optimization (PPO) Schulman et al. (2017) and its variations to both diffusion and consistency models Oertell et al. (2024); Wallace et al. (2023); Black et al. (2024); Fan et al. (2023); Yang et al. (2024), our work emphasizes that such complex training approaches may not always be necessary. We show that using the *reparameterization trick* Kingma & Welling (2014), we can directly optimize the regularized RLHF objective by backpropagating through the entire generation trajectory. Our experiments consistently demonstrate that direct optimization of the RLHF objective can outperform the use of PPO both in training stability and efficiency. Furthermore, a user study corroborates the effectiveness of our approach, underscoring its potential as a simpler yet robust alternative for training consistency models. Our contributions in this work can be summarized as follows:

- We reformulate the RLHF optimization problem as a direct optimization objective by leveraging the reparameterization trick, allowing efficient backpropagation through the generation trajectory. This reformulation transforms a zero-order optimization problem into a first-order one, significantly enhancing optimization efficiency. Empirically, our results demonstrate that this approach achieves performance on par with or superior to policy gradient based methods while requiring substantially less hyperparameter tuning and enabling faster training.

- We formulate and analyze the role of distributional regularization in RLHF for fine-tuning consistency models, demonstrating its impact on training stability, efficiency and reward alignment. The benefit of distributional regularization is also theoretically proved in the Supplementary material A.1.

- We conduct a comprehensive empirical analysis of various $f$-divergence measures for regularization, highlighting their influence on training stability and model performance.

## 2 Preliminaries

**Consistency Models.** Diffusion models define a family of probability distributions $p_t(x)$ parameterized by time $t \in [0, T]$, where a clean data sample $x_0 \sim p_0(x)$ undergoes a gradual noising process. At the terminal timestep $T$, the data distribution converges to an isotropic Gaussian prior, i.e., $x_T \sim \mathcal{N}(0, I)$. The forward diffusion process follows the transition kernel:

$$q_t(x_t|x_0) = \mathcal{N}(x_t; \alpha_t x_0, \sigma_t I), \tag{1}$$

where $\alpha_t$ and $\sigma_t$ govern the noise schedule at each step.

This stochastic process can equivalently be described by the following Stochastic Differential Equation (SDE) Song et al. (2021); Lu et al. (2023); Karras et al. (2022):

$$dx_t = f(t)x_t + g(t)dw_t, \tag{2}$$

where $w_t$ denotes standard Brownian motion, and $f(t)$, $g(t)$ are functions determined by the noise schedule $\alpha_t, \sigma_t$. This SDE defines a stochastic process $x_t$ whose marginals are $q_t(x)$. The corresponding reverse-time dynamics depend on the score function $\nabla_x \log q_t(x)$, and can be reformulated as an Ordinary Differential Equation (ODE) Song et al. (2021):

$$\frac{dx_t}{dt} = f(t)x_t + \frac{1}{2}g^2(t)\epsilon_\theta(x_t, t). \tag{3}$$

Here, $x_T \sim \mathcal{N}(0, I)$, and $\epsilon_\theta$ is a neural network trained to approximate the score function $\nabla_x \log q_t(x_t)$.

To enhance conditional generation, classifier-free guidance Ho & Salimans (2022) modifies the predicted noise as:

$$\hat{\epsilon}_\theta(x_t, \omega, c, t) = (1 + \omega)\epsilon_\theta(x_t, c, t) - \omega\epsilon_\theta(x_t, \phi, t), \tag{4}$$

where $c$ is a conditioning input (e.g., a text prompt), and $\omega$ is the guidance scale controlling the trade-off between diversity and specificity. Substituting Eq. 4 into Eq. 3 yields the Augmented Probability Flow ODE (APFODE) Luo et al. (2023):

$$\frac{dx_t}{dt} = f(t)x_t + \frac{1}{2}g^2(t)\hat{\epsilon}_\theta(x_t, \omega, c, t). \tag{5}$$

Consistency models aim to accelerate diffusion-based generation by learning a direct mapping from noisy samples to clean outputs in one or a few inference steps. Instead of solving Eq. 3 iteratively, a consistency model approximates the entire ODE trajectory directly. Specifically, given two time steps $t' > t$, if $x_{t'} \sim p_{t'}(x)$, integrating Eq. 3 yields $x_t \sim p_t(x)$, and ultimately $x_0 \sim p_0(x)$. The consistency model, parameterized by $f_\theta$, learns:

$$f_\theta(x_t, t) = f_\theta(x_{t'}, t') = x_0, \tag{6}$$

where $x_0$ denotes the recovered sample corresponding to time 0. Training enforces this consistency by minimizing a distance function $d(f_\theta(x_t, t), f_\theta(x_{t'}, t'))$, typically using the $L_2$ norm Song et al. (2023).

Following prior work Song et al. (2023), $f_\theta$ is parameterized as:

$$f_\theta(x_t, t) = c_{\text{skip}}(t)x_t + c_{\text{out}}(t)F_\theta(x_t, t), \tag{7}$$

where $c_{\text{skip}}(t)$ and $c_{\text{out}}(t)$ are differentiable functions with constraints $c_{\text{skip}}(0) = 1$ and $c_{\text{out}}(0) = 0$. $F_\theta$ denotes a neural network with learnable parameters $\theta$. Incorporating classifier-free guidance, we write $f_\theta(x_{t_k}, \omega, c, t_k)$ to directly predict the solution to Eq. 5.

A key advantage of consistency models lies in their ability to generate high-quality samples with few inference steps. The probability flow trajectory is discretized into $K$ decreasing timesteps $T = t_K > t_{K-1} > \cdots > t_1 = 0$, where each step refines the sample toward $x_0$. This defines a generative policy $\pi_\theta$ that maps noisy inputs to high-fidelity outputs efficiently.

Using Eq. 1, any sample at time $t_n$ can be written as $x_{t_n} = \alpha_{t_n}x_0 + \sigma_{t_n}z$, $z \sim \mathcal{N}(0, I)$. Given a sample $x_k$ at time $t_k$, we approximate $x_0$ with a consistency function $\tilde{x}_k = f_\theta(x_k, \omega, c, t_k)$. The next-step sample is computed as:

$$x_{k-1} = \alpha_{t_{k-1}}\tilde{x}_k + \sigma_{t_{k-1}}\epsilon_{k-1}, \quad \epsilon_{k-1} \sim \mathcal{N}(0, I).$$

At the final step, $x_0 \approx \tilde{x}_1 = f_\theta(x_1, \omega, c, t_1)$, since $\alpha_0 = 1$ and $\sigma_0 = 0$.

**Regularized RLHF.** Reinforcement Learning from Human Feedback (RLHF) aligns generative model outputs with human preferences by optimizing a reward signal derived from feedback. In the regularized form, the objective is:

$$\mathcal{L}_{\text{RLHF}} = \mathbb{E}_{\tau \sim \pi_\theta}[R(\tau)] + \beta \mathcal{D}(\pi_\theta \| \pi_{\theta_{\text{ref}}}), \tag{8}$$

where $\tau$ denotes a trajectory sampled from the generation policy $\pi_\theta$ (defined by Algorithm 1), $R(\tau)$ is the trajectory reward (typically computed at the final generation step), $\pi_{\theta_{\text{ref}}}$ is the reference policy corresponding to the pretrained generative model (which we refer to as the base model), and $\beta$ controls regularization strength.

The reward model $R(\tau)$ encodes human preferences by assigning higher values to outputs that better align with human judgments. In practice, reward models are learned from human preference data —forexample,PickScore Kirstain et al. (2023) and HPSv2 Wu et al. (2023) are CLIP-based models trained on human pairwise comparisons of generated images. These reward models are differentiable with respect to the generated image, making them compatible with our first-order optimization approach. Non-differentiable rewards, such as compressibility or human-only feedback, are not directly applicable to our method.

The regularization term prevents overfitting to the reward model and stabilizes optimization, analogous to its role in large language model fine-tuning Ziegler et al. (2019); Stiennon et al. (2020). Common choices for the divergence $\mathcal{D}$ include the Kullback–Leibler (KL) divergence, Jensen–Shannon (JS) divergence, Hellinger distance, and Fisher divergence—each influencing learning dynamics differently.

*f*-**Divergence.** The $f$-divergence is a general framework that unifies many divergence measures. Given a convex function $f(x) : \mathbb{R}^+ \to \mathbb{R}$ satisfying $f(1) = 0$, the $f$-divergence between distributions $p_1$ and $p_2$ over a common space $\mathcal{X}$ is defined as Liese & Vajda (2006):

$$\mathcal{D}_f(p_1 || p_2) = \mathbb{E}_{x \sim p_2}\left[f\left(\frac{p_1(x)}{p_2(x)}\right)\right]. \tag{9}$$

Different choices of $f(x)$ recover well-known divergences such as KL, JS, Fisher, and Hellinger, providing flexibility in regularizing policies.

## 3 Related Works

**Diffusion Models:** Diffusion models have emerged as a powerful class of generative models for tasks involving modeling of continuous data distributions. Inspired by non-equilibrium thermodynamics, these models learn to reverse a stochastic process that gradually adds noise to data, effectively learning the data distribution by reversing this process during generation Sohl-Dickstein et al. (2015); Ho et al. (2020); Song et al. (2021); Song & Ermon (2020). The iterative nature of diffusion models, where samples are generated through a sequence of denoising steps, allows them to produce high-quality outputs. However, this multi-step generation process is computationally intensive, leading to long inference times. To address this, recent work has focused on developing more efficient variants, such as DDIM Song et al. (2022) & DEIS Zhang & Chen (2023), which accelerates sampling by reducing the number of steps while maintaining output quality. Despite these advancements, the slow sampling speed of diffusion models remains a significant limitation, particularly when integrated with reinforcement learning frameworks for fine-tuning, as storing gradients for every timestep remains memory-intensive, even when only fine-tuning the LoRA layers Hu et al. (2021).

**Consistency Models:** Consistency models present an alternative approach to generative modeling by enabling single-step or few-step generation Song et al. (2023). These models are trained to maintain consistency in their outputs across multiple forward passes, facilitating much faster sampling compared to traditional diffusion models. The core idea is to train a network capable of directly mapping noise from any point in time to the target data distribution in a single step. For multi-step generation, noise is added at each step to the predicted target distribution sample, followed by the re-application of the denoising network, which allows these models to achieve competitive results in a limited number of steps. This approach drastically reduces the computational overhead during inference, making it particularly advantageous for scenarios requiring rapid generation. The ability of consistency models to produce high-quality samples in just a few steps also makes them well-suited for integration with reinforcement learning frameworks, where efficient feedback is crucial. For instance, RLCM Oertell et al. (2024) employs PPO to fine-tune a consistency model. While their work is closely related to ours, the key distinction lies in our use of direct reward optimization instead of PPO; moreover our objective focuses on optimizing a regularized version of the reward signal.

**Reinforcement Learning from Human Feedback:** RLHF has gained traction in aligning generative models with human preferences, particularly in cases where explicit reward signals are sparse or difficult to define Christiano et al. (2023). By leveraging human feedback, RLHF helps models generate outputs that better match human expectations. In generative modeling, RLHF fine-tunes models using reward signals derived from human judgments, improving output quality and relevance. However, integrating RLHF with

diffusion models presents challenges due to slow sampling, memory constraints, and sparse reward signals, typically provided only at the end of the generation process. Various adaptations, such as PPO and its variants, have been explored to mitigate these issues, but they require complex training procedures with extensive hyperparameter tuning. End-to-end RLHF training methods, like DRaFT Clark et al. (2024), employ techniques such as gradient checkpointing and truncated backpropagation to manage computational overhead. Meanwhile, approaches based on Direct Preference Optimization (DPO) Yang et al. (2024); Wallace et al. (2023); Liang et al. (2024) approximate the training objective to decouple it from the generation steps, allowing optimization without storing per-step gradients. RLHF has also been utilized to enhance generation diversity Shekhar et al. (2024), though this is not the focus of our work.

## 4 Methodology

Since our generation process involves multiple intermediate steps—analogous to the reasoning chain in LLMs where conditional regularization is applied at each step—we aggregate divergences over conditional distributions $p(x_{k-1}|x_k, c)$ for $k = 2, \ldots, K$. According to Algorithm 1, each conditional distribution is Gaussian:

$$p_k(\cdot|\theta, x_k, c) = \mathcal{N}(\alpha_{t_{k-1}} f_{\theta_{\mathrm{ref}}}(x_k, \omega, c, t_k), \sigma^2_{t_{k-1}}). \tag{10}$$

The overall regularization can thus be written as:

$$\mathcal{D}(\pi_\theta \| \pi_{\theta_{\mathrm{ref}}}) = \mathbb{E}_{\tau \sim \pi_\theta} \sum_{k=2}^{K} \mathcal{D}_f(p_k(\cdot|\theta, x_k, c) \| p_k(\cdot|\theta_{\mathrm{ref}}, x_k, c)),$$

where $\mathcal{D}_f$ is a chosen $f$-divergence. Importantly, this regularization is reparameterizable.

In standard RL, the gradient of $\mathbb{E}_{\tau \sim \pi_\theta}[R(\tau)]$ w.r.t. $\theta$ is estimated via policy gradients or PPO Schulman et al. (2017). Instead, we adopt a reparameterization approach, which reduces variance and allows gradient flow through the entire generative process—similar to VAEs Kingma & Welling (2014). The randomness in Algorithm 1 arises solely from the Gaussian variables $\epsilon_K, \ldots, \epsilon_1$, which can be aggregated as a single variable $\epsilon$. Thus, the trajectory $\tau = G(\theta, \epsilon, c) = \{(x_k, \tilde{x}_k)\}_{k=1}^{K}$, where $x_k = G_k(\theta, \epsilon, c)$ and $\tilde{x}_k = f_\theta(x_k, \omega, c, t_k)$.

The reparameterized RLHF objective becomes:

$$\mathcal{L}_{\mathrm{RLHF}} = \mathbb{E}_{\epsilon \sim \mathcal{N}(0,I)} \Bigg[ R(G_0(\theta, \epsilon, c), c) + \beta \sum_{k=2}^{K} \tag{11}$$

$$D_f(p_k(\cdot|\theta, G_k(\theta, \epsilon, c), c) \| p_k(\cdot|\theta_{\mathrm{ref}}, G_k(\theta, \epsilon, c), c)) \Bigg].$$

We refer to Eq. 11 as the *direct optimization formulation* for RLHF. Gradients can be computed via backpropagation through the generation process in Algorithm 1. Since all conditional distributions are Gaussian, many divergences admit closed-form computation (see Table 3). For divergences without closed forms, such as JS, reparameterization can again be applied using Eq. 9. The resulting training algorithm is summarized in Algorithm 2.

Finally, the reward $R(\tau)$ encodes human preferences, typically depending on the final image $x_0$ and condition $c$, and is assumed differentiable w.r.t. $x_0$. The regularization coefficient $\beta$ balances the reward and regularization terms; empirically, we find stable learning when the divergence term is scaled to be an order of magnitude smaller than the rewards.

## 5 Experiments

**Datasets:** To train our models in an online fashion—where each model is trained exclusively on its own generated data while being updated iteratively—we utilized 4,000 text prompts (without images) randomly

| Method | HPSv2 | | | PickScore | | |
|---|---|---|---|---|---|---|
| | PickScore | Aesthetic Score | ImageReward | HPSv2 | Aesthetic Score | ImageReward |
| No Regularization | 21.626 ± 0.072 (0.43% ↑) | **6.489 ± 0.056** (4.04% ↑) | 0.471 ± 0.086 (23.95% ↑) | 2.92 ± 0.01 (4.29% ↑) | 6.323 ± 0.023 (1.38% ↑) | 0.409 ± 0.145 (7.63% ↑) |
| JS-Divergence | 21.774 ± 0.137 (1.12% ↑) | 6.469 ± 0.096 (3.72% ↑) | 0.685 ± 0.48 (80.26% ↑) | 2.97 ± 0.08 (6.07% ↑) | **6.414 ± 0.591** (2.84% ↑) | 0.485 ± 0.066 (27.63% ↑) |
| KL-Divergence | **21.958 ± 0.055** (1.97% ↑) | 6.452 ± 0.036 (3.45% ↑) | 0.652 ± 0.060 (71.58% ↑) | **2.97 ± 0.10** (6.07% ↑) | 6.358 ± 0.302 (1.94% ↑) | **0.517 ± 0.497** (36.05% ↑) |
| Hellinger | 21.859 ± 0.039 (1.51% ↑) | 6.396 ± 0.085 (2.55% ↑) | **0.779 ± 0.073** (105.00% ↑) | 2.82 ± 0.03 (0.71% ↑) | 6.293 ± 0.028 (0.90% ↑) | 0.402 ± 0.059 (5.79% ↑) |
| Fisher Divergence | 21.774 ± 0.075 (1.12% ↑) | 6.380 ± 0.058 (2.29% ↑) | 0.551 ± 0.072 (45.00% ↑) | 2.88 ± 0.02 (2.86% ↑) | 6.339 ± 0.047 (1.64% ↑) | 0.435 ± 0.068 (14.47% ↑) |
| RLCM | 21.538 ± 0.034 (0.02% ↑) | 6.347 ± 0.054 (1.76% ↑) | 0.584 ± 0.042 (53.68% ↑) | 2.73 ± 0.08 (2.50% ↓) | 6.248 ± 0.01 (0.18% ↑) | 0.298 ± 0.066 (21.58% ↓) |
| DDPO | 21.654 ± 0.018 (0.57% ↓) | 6.177 ± 0.028 (0.55% ↑) | 0.747 ± 0.051 (12.22% ↓) | 2.98 ± 0.01 (2.93% ↓) | 6.052 ± 0.117 (1.48% ↓) | 0.685 ± 0.029 (19.51% ↓) |
| D3PO | 21.670 ± 0.055 (0.50% ↑) | 6.230 ± 0.003 (1.42% ↓) | 0.809 ± 0.040 (4.94% ↓) | 2.79 ± 0.013 (9.12% ↓) | 6.052 ± 0.117 (1.48% ↓) | 0.559 ± 0.104 (34.31% ↓) |
| DPOK | 21.701 ± 0.014 (0.35% ↓) | 6.189 ± 0.031 (0.75% ↑) | 0.791 ± 0.031 (7.05% ↓) | 3.01 ± 0.01 (1.95% ↓) | 6.071 ± 0.0025 (1.17% ↓) | 0.713 ± 0.037 (16.22% ↓) |
| DRaFT | 21.497 ± 0.158 (1.29% ↓) | 6.122 ± 0.045 (0.34% ↓) | 0.799 ± 0.009 (6.11% ↓) | 3.03 ± 0.02 (1.30% ↓) | 6.163 ± 0.034 (0.33% ↑) | 0.781 ± 0.013 (8.23% ↓) |

| Method | ImageReward | | | Aesthetic Score | | |
|---|---|---|---|---|---|---|
| | PickScore | Aesthetic Score | HPSv2 | PickScore | HPSv2 | ImageReward |
| No Regularization | 21.559 ± 0.029 (0.12% ↑) | 6.258 ± 0.029 (0.34% ↑) | 2.72 ± 0.07 (2.86% ↓) | 21.529 ± 0.084 (0.02% ↓) | 2.88 ± 0.07 (2.86% ↑) | 0.375 ± 0.069 (1.32% ↓) |
| JS-Divergence | **21.752 ± 0.053** (1.02% ↑) | **6.436 ± 0.138** (3.19% ↑) | **2.96 ± 0.08** (5.71% ↑) | 21.550 ± 0.064 (0.08% ↑) | 2.89 ± 0.08 (3.21% ↑) | 0.435 ± 0.037 (14.47% ↑) |
| KL-Divergence | 21.751 ± 0.039 (1.01% ↑) | 6.324 ± 0.043 (1.39% ↑) | 2.95 ± 0.06 (5.36% ↑) | **21.801 ± 0.03** (1.24% ↑) | **2.99 ± 0.07** (6.79% ↑) | **0.536 ± 0.079** (41.05% ↑) |
| Hellinger | 21.684 ± 0.025 (0.70% ↑) | 6.301 ± 0.0299 (1.03% ↑) | 2.89 ± 0.04 (3.21% ↑) | 21.692 ± 0.103 (0.74% ↑) | 2.94 ± 0.03 (5.00% ↑) | 0.532 ± 0.059 (40.00% ↑) |
| Fisher Divergence | 21.744 ± 0.050 (0.98% ↑) | 6.361 ± 0.046 (1.99% ↑) | 2.90 ± 0.06 (3.57% ↑) | 21.743 ± 0.048 (0.98% ↑) | 2.98 ± 0.05 (6.43% ↑) | 0.531 ± 0.040 (39.74% ↑) |
| RLCM | 21.670 ± 0.05 (0.64% ↑) | 6.285 ± 0.058 (0.77% ↑) | 2.89 ± 0.08 (3.21% ↑) | 21.675 ± 0.071 (0.66% ↑) | 2.89 ± 0.02 (3.21% ↑) | 0.407 ± 0.081 (7.11% ↑) |
| DDPO | 21.553 ± 0.087 (1.03% ↓) | 6.117 ± 0.019 (0.42% ↓) | 3.03 ± 0.01 (1.30% ↓) | 21.564 ± 0.085 (0.98% ↓) | 3.02 ± 0.03 (1.63% ↓) | 0.740 ± 0.013 (13.04% ↓) |
| D3PO | 21.545 ± 0.122 (1.07% ↓) | 6.141 ± 0.048 (0.03% ↓) | 3.02 ± 0.08 (1.63% ↓) | 21.485 ± 0.05 (1.35% ↓) | 3.03 ± 0.03 (1.30% ↓) | 0.592 ± 0.01 (30.43% ↓) |
| DPOK | 21.602 ± 0.063 (0.81% ↓) | 6.125 ± 0.020 (0.29% ↓) | 3.02 ± 0.01 (1.63% ↓) | 21.656 ± 0.019 (0.56% ↓) | 3.02 ± 0.02 (1.63% ↓) | 0.764 ± 0.015 (10.22% ↓) |
| DRaFT | 21.692 ± 0.021 (0.39% ↓) | 6.112 ± 0.018 (0.50% ↓) | 3.01 ± 0.02 (1.95% ↓) | 21.613 ± 0.020 (0.76% ↓) | 3.04 ± 0.05 (0.98% ↓) | 0.833 ± 0.026 (2.12% ↓) |

Table 1: **Comparison of different regularization techniques across multiple reward models:** Each model is trained separately using PickScore Kirstain et al. (2023), HPSv2 Wu et al. (2023), ImageReward Xu et al. (2023), and Aesthetic Score Wang et al. (2022). Models are not evaluated on the reward function they were trained on to avoid bias. We also show in brackets the relative performance change from the base model.

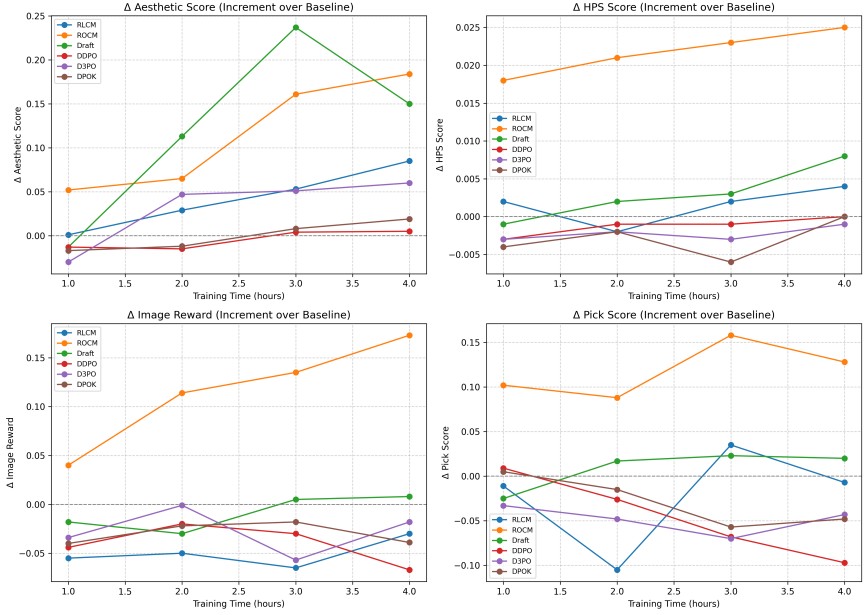

Figure 2: **Training Efficiency:** This figure shows the relative improvement in validation reward scores over training time for each method. Our method consistently achieves faster gains across all reward models, indicating improved data efficiency. While improvements for the Aesthetic Score reward model are initially marginal, the other reward models clearly demonstrate that ROCM converges more quickly, achieving higher scores with fewer training steps. We can also see that DRaFT usually performs better than other policy gradient-based diffusion approaches and RLCM also performing better than other diffusion methods

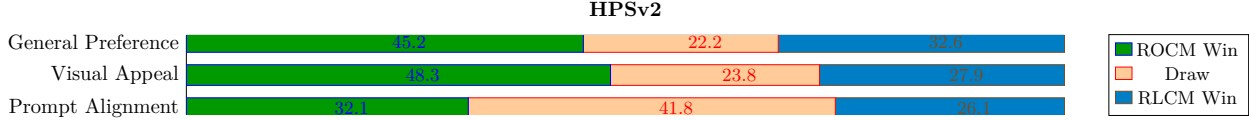

**HPSv2**

Figure 3: **User Study:** We compare our best-performing models trained with each reward function to RLCM Oertell et al. (2024), with all models fine-tuned on their respective reward models. Following the protocol of SPO Liang et al. (2024), we conduct a user study using 300 prompts, sampled from PartiPrompts Yu et al. (2022) and HPS Wu et al. (2023) in a 1:2 ratio. The results here focus on the HPSv2 reward model, while outcomes for other reward models are presented in Fig: 10 & 11.

sampled from the Pick-a-Pic V1 dataset[1], as employed in Liang et al. (2024). This prompt dataset was used to fine-tune models with PickScore Kirstain et al. (2023), HPSv2 Wu et al. (2023), ImageReward Xu et al. (2023), and Aesthetic Score Wang et al. (2022). Furthermore, for generating images in Fig: 8, we trained models using Aesthetic Score on a smaller set of 45 animal-related prompts, as in Black et al. (2024).

For quantitative evaluation, we report results on 500 validation prompts present in the `validation_unique` split of the Pick-a-Pic V1 dataset Kirstain et al. (2023), which was also utilized in Liang et al. (2024). We train five models following Algorithm 2, each incorporating a different regularization method: No regularization, KL-Divergence, JS-Divergence, Hellinger Distance, and Fisher Divergence. Our models are compared against baseline methods, including RLCM Oertell et al. (2024), DDPO Black et al. (2024), DPOK Fan et al. (2023), D3PO Yang et al. (2024) and DRaFT Clark et al. (2024). Specifically, RLCM applies PPO to consistency models, DDPO employs PPO for diffusion models, DPOK utilizes policy gradient with a KL-regularized reward, D3PO extends DPO Rafailov et al. (2024) to diffusion models, and DRaFT applies direct reward optimization to diffusion models Clark et al. (2024).

**Implementation Details:** In our experiments, we employ an 8-step consistency model and a 20-step diffusion model, both utilizing classifier-free guidance Ho & Salimans (2022) with a guidance scale of $\omega = 7.5$. For diffusion-based and consistency-based methods, we use Dreamshaper v7[2], a fine-tuned version of Stable Diffusion v1.5, along with its corresponding consistency model counterpart Luo et al. (2023) as base models respectively. These are further fine-tuned with trainable LoRA Hu et al. (2021) layers. For consistency-based methods, we set the LoRA rank and scaling factor $\alpha$ to 8, as higher values led to overfitting in RLCM and offered minimal gains in our setting. For diffusion-based methods like DPOK, DDPO, and D3PO, we adopted the de-

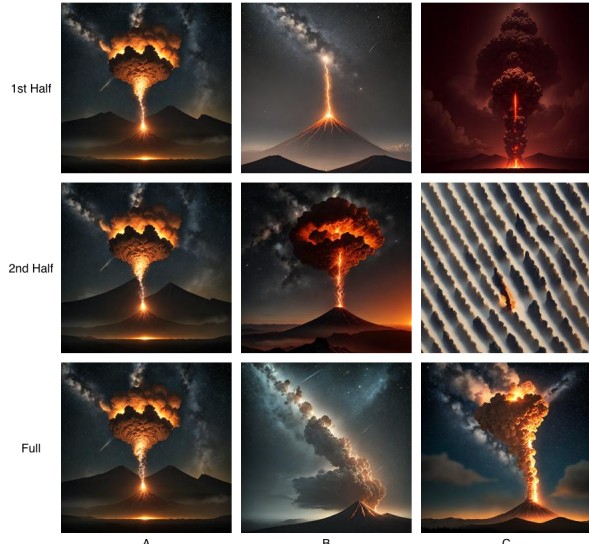

Figure 4: **Comparison of regularization strategies across the diffusion chain.** The first two rows show generations when only the **first half** or **second half** is regularized, and the last row shows **full-chain regularization**. Columns A, B, and C correspond to training stages: A = 100 steps, B = 8000 steps, and C = 15000 steps. Beyond C, the **second-half-regularized** model exhibits clear mode collapse, so visualization is stopped. First-half regularization preserves global structure even under over-fitting but yields lower perceptual quality. Second-half regularization produces sharper, more detailed images but overfits rapidly, leading to unrealistic generations. Full-chain regularization retains the visual fidelity of second-half regularization while preventing early over-fitting, combining the strengths of both.

---

[1]https://huggingface.co/datasets/yuvalkirstain/PickaPic
[2]https://huggingface.co/Lykon/dreamshaper-7

fault parameters and implementation provided by D3PO[3] Yang et al. (2024). Since there is no publicly available implementation of DRaFT, we performed a hyperparameter search to obtain the best set of parameters. To ensure fair comparison, we used the same learning rate across all models. Further experimental details can be found in Appendix A.3.

We explore multiple $f$-divergences, namely KL-Divergence, Fisher Divergence, Hellinger Distance, and JS Divergence, incorporating a hyperparameter $\beta$ to regulate regularization strength. The optimal values for these hyperparameters are detailed in the appendix (A.3). For all divergences except Jensen-Shannon, we utilize the closed-form solutions provided in Table 3. Since JS Divergence lacks a closed-form solution, we resort to sampling for its computation.

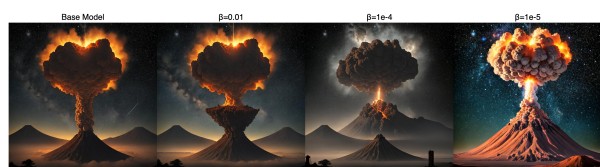

Figure 5: **Role of $\beta$ in Model Deviation:** As $\beta$ decreases, the generated images deviate more from the base model. Moderate values of $\beta$ yield outputs that better align with the reward model and exhibit higher quality, while very low values lead to significant deviation and potential overfitting.

**Evaluation Metrics:** For evaluation, we use 4 automated metrics: PickScore Kirstain et al. (2023), HPSv2 Wu et al. (2023), Aesthetic Score Wang et al. (2022), and ImageReward Xu et al. (2023). All metrics, except Aesthetic Score, are prompt-aware, while Aesthetic Score is prompt-agnostic and evaluates only the aesthetic quality using a regression head on top of the CLIP vision encoder. Each reward model has been trained on human preference data to approximate human image quality judgments. PickScore and HPSv2 utilize a CLIP-based model trained on human preferences related to aesthetic quality and prompt-to-image alignment while ImageReward uses a BLIP-based model for

the same. Particularly for each model trained on a reward model we exclude evaluation on the same model to avoid bias and give a broder evaluation. Beyond automated metrics, we conduct a user study similar to Liang et al. (2024). We recruited 10 participants to evaluate 300 image pairs generated by RLCM and our best-performing models for each reward model. The prompts for this study are randomly sampled from a mixture of the PartiPrompts dataset Yu et al. (2022) and the HPSv2 dataset Wu et al. (2023), maintaining a 1:2 ratio.

## 5.1 Results

We evaluate all regularization methods across different reward models using the previously described evaluation metrics. The results are summarized in Table 1, with each table corresponding to models trained on a specific reward model. We report averaged results with standard deviation across three runs using different seeds, after 10,000 training steps for each method. Additionally, we present training time vs. performance graphs in Fig. 2. Across all tables and metrics, regularized-ROCM consistently matches or outperforms competing methods in automatic evaluations. Even the non-regularized variant often performs comparably or better than baselines. Both regularized-ROCM and RLCM surpass their diffusion-based counterparts on most metrics, underscoring the benefits of consistency models, which are easier to fine-tune than diffusion models due to shorter trajectories and fewer issues with sparse rewards. As shown in Fig. 2, regularized-ROCM also achieves superior training efficiency compared to RLCM, likely because PPO relies on noisy zeroth-order gradients, while our method leverages more stable first-order updates. We also see that DRaFT performed better than other diffusion approaches on average showing that direct reward optimization is more stable than policy gradient approaches. Table 1 further shows that diffusion-based models often underperform compared to their base model, typically overfitting to the reward model and sacrificing general performance—a trend also seen in RLCM and non-regularized ROCM. Regularized-ROCM avoids this issue, indicating that both regularization and first-order optimization are key to robust performance.

The user study in Fig. 3 & 10 supports these findings, with our methods significantly outperforming RLCM across all reward models in terms of *Visual Appeal* and *General Preference*. Gains in *Prompt Alignment* are

---

[3]Available at: `https://github.com/yk7333/d3po/tree/main/scripts/rm`

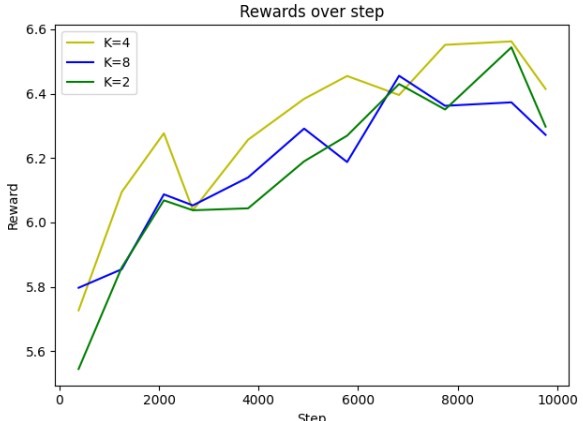 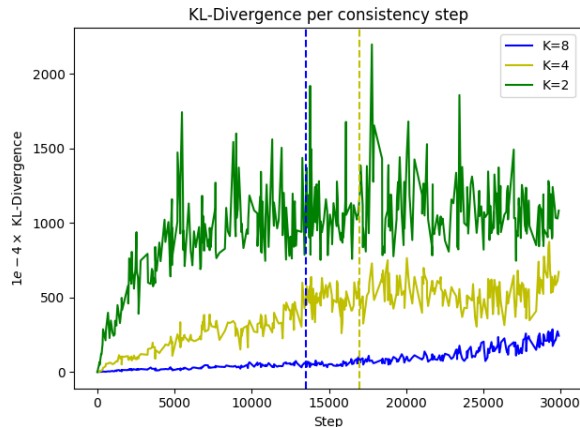

(a) **Effect of Number of Consistency Steps K on Training:** We plot reward as a function of training time for K $\in \{2, 4, 8\}$. Across all settings, ROCM converges to comparable final reward levels within similar training budgets, demonstrating robustness to the choice of K over the practical operating range of consistency models.

(b) **KL divergence across training steps for different consistency step counts.** Models with fewer steps ($K = 2$) show higher per-step divergence, indicating stronger deviation from the reference model, =while larger $K$ (e.g., 8) maintains smoother and more stable updates.

smaller, especially for Aesthetic Score, which is prompt-agnostic. In contrast, prompt-aware reward models lead to better alignment.

## 5.2 Further Analysis

**Effect of Regularization Strength ($\beta$):** Fig. 7b and Fig. 5 illustrate how the regularization strength $\beta$ influences model performance. In Fig. 7b, we report results after 10,000 iterations using KL-Divergence for regularization. At higher values of $\beta$, human preference and reward model (RM) predictions remain closely aligned, indicating that strong regularization keeps the model near the initial distribution and prevents reward overoptimization. As $\beta$ decreases, the RM's predicted preference rises faster than the actual human preference. At $\beta = 10^{-4}$, human preference peaks and then declines, while RM predictions continue increasing—an indicator of reward hacking, where RM scores inflate despite noisier outputs. Similar behavior has been noted in Stiennon et al. (2022) for large language models. Fig. 5 shows a corresponding drop in visual quality at lower $\beta$ values, with artifacts or degraded images appearing despite higher RM scores. In contrast, larger $\beta$ values produce more stable and coherent generations, though with slower reward gains. This highlights the need to carefully tune $\beta$ to balance stability, alignment, and genuine quality improvement.

**Effectiveness of reward models:** From Table 1 and Fig. 2, we observe that training with ImageReward, PickScore, and HPSv2 as reward signals consistently leads to substantial improvements in both generation quality and prompt alignment. Among these, PickScore demonstrates notably lower sensitivity to image quality variations, resulting in a weaker learning signal and therefore more limited improvements compared to the other reward models. Models trained with Aesthetic Score exhibit strong gains in visual quality; however, the improvements in prompt alignment remain modest. This is expected, as Aesthetic Score is inherently prompt-agnostic and therefore can encourage models to generate visually pleasing but repetitive images that drift away from the input prompt. Finally, across all reward models, our method shows a consistently steeper reward-increment–versus–time slope, indicating greater training efficiency and faster convergence compared to baseline approaches.

**Effect of regularization position:** We study the effect of applying KL-regularization at different positions in the consistency generation process when using the aesthetic score as the reward model. The process has 8 steps, and we compare two variants: (i) regularizing only the first 4 steps, and (ii) regularizing only the last 4 steps. As shown in Figure 7a, omitting regularization in the latter half causes rapid overfitting, while applying it only in the later steps leads to more stable training. Both settings, however, degrade earlier than

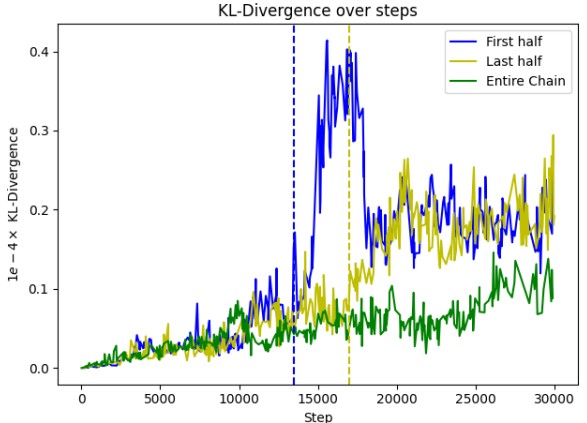

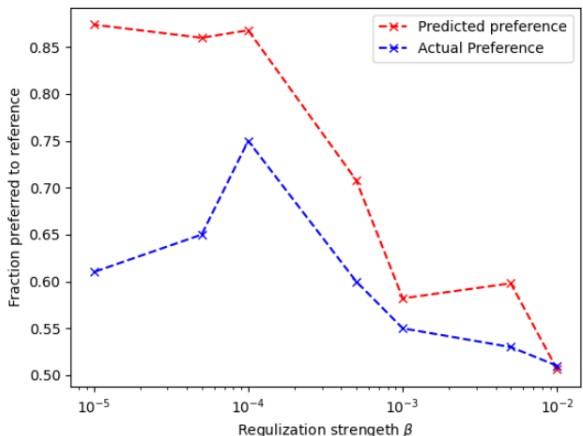

(a) **Effect of regularization position.** KL-divergence over training steps when KL-regularization is applied to different portions of the consistency generation process. Applying KL only in the first half leads to rapid overfitting (blue), while applying it only in the last half improves stability (yellow). Both settings, however, degrade earlier than applying KL across the entire chain (green).

(b) **Effect of $\beta$:** We can see as $\beta$ decreases, we observe an initial improvement in model performance. However, with further reduction in $\beta$, the actual preference reaches a peak and then begins to decline, indicating reward hacking.

applying KL across the entire chain. Since later steps are known to focus more on generating fine details, regularizing them prevents the model from overfitting to noisy signals in the reward model. We can see the qualitatively effects of this in Fig. 4. This indicates that reward hacking arises primarily from the model fitting to fine-detail noise, and that later-step regularization is critical for mitigating this issue.

**Effect of the number of consistency steps $K$.** We analyze how the number of consistency inference steps $K$ influences the stability and regularization behavior of our model. Fig. 6a & 6b plots the per-step KL divergence between the fine-tuned and reference models for different $K$ and also the reward per step. We observe that models trained with fewer steps (e.g., $K = 2$) exhibit substantially higher KL divergence compared to those trained with larger $K$ (e.g., $K = 8$). This occurs because shorter trajectories require each consistency step to make larger updates to approximate the full denoising process, thereby deviating more strongly from the reference distribution. In contrast, larger $K$ distributes the transformation across more intermediate steps, resulting in smoother, smaller per-step deviations and better regularization. Consequently, while smaller $K$ can yield faster inference, it also tends to amplify reward-driven deviations—potentially increasing the risk of reward hacking. For the KL divergence comparison in Figure 6b, we fix $\beta = 10^{-4}$ across all values of $K$ to isolate the effect of $K$ alone. For Figure 6a, we instead tune $\beta$ appropriately for each $K$—using a larger $\beta$ for smaller $K$ and a smaller $\beta$ for larger $K$, reflecting the higher per-step divergence induced by shorter trajectories. Crucially, we observe that with this adjustment, the final reward remains consistent across all values of $K$, demonstrating that the choice of $K$ does not fundamentally limit model performance as long as $\beta$ is set accordingly.

# 6 Conclusions & Limitations

We presented a simple yet effective method for fine-tuning consistency models using direct reward optimization via the reparameterization trick. Our approach bypasses the complexity of PPO-style optimization, enabling stable training by propagating gradients across the entire generation trajectory. We showed that distributional regularization—penalizing deviations from the initial model—improves reward alignment, sample quality, and robustness to reward hacking. Across multiple divergence measures, regularization also enhanced generalization and training efficiency over unregularized baselines.

While consistency models offer significant efficiency advantages in the RLHF fine-tuning regime, they are not yet the dominant paradigm in generative modeling more broadly. Their training dynamics are more sensitive than those of standard diffusion models, and effective training typically requires a parent diffusion model for initialization or distillation, making them dependent on diffusion model infrastructure rather than being fully standalone. Additionally, the ecosystem of pretrained consistency model checkpoints and associated tooling remains considerably less mature, and a quality gap persists relative to diffusion models at high step counts. Our work does not claim to resolve these limitations; rather, it demonstrates that *within* the RLHF fine-tuning regime, the shorter generation horizon of consistency models is a concrete and exploitable advantage.

A key limitation is that our method requires differentiable reward functions, limiting applicability in settings with non-differentiable objectives such as entropy, compression metrics, or human preference-only feedback, where policy-gradient or other gradient-free methods remain necessary.

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

## A Appendix

### A.1 Theoretical Analysis

In practical applications of reinforcement learning, the reward model $R(\tau)$ is often learned from the data, which will not be the same as the ground truth reward, which we will denote as $R_*(\tau)$. Here $\tau$ denotes a trajectory generated by the underlying generation policy $\pi_\theta$. The error becomes larger for out of distribution data. To mathematically quantify the difference, we make the following assumption, where we may assume that in-distribution data correspond to trajectory generated from the reference policy $\pi_{\theta_{\text{ref}}}$. For simplicity, in the following, we consider powerful generation policy models that may contain all policies, so that we will simply omit the corresponding model parameter $\theta$. Therefore we denote generation policies simply by $\pi$, and reference policy by $\pi_{\text{ref}}$.

**Assumption 1.** *We assume that the in distribution (id) error of reward model is*

$$\mathbb{E}_{\tau \sim \pi_{\text{ref}}}(R(\tau) - R_*(\tau))^2 = \epsilon_{\text{id}}^2,$$

*and the worst case out of distribution error is*

$$\sup_\tau |R(\tau) - R_*(\tau)|^2 = \epsilon_{\text{ood}}^2.$$

In typical scenario, in distribution error can be significantly smaller than out of distribution error. That is it is natural to assume that

$$\epsilon_{\text{id}} \ll \epsilon_{\text{ood}}.$$

In the following, we will show theoretical that without distributional regularization, in the worst case, there are examples such that the performance of learned policy can overfit the out distribution error, leading to

performance worse even than the reference policy. This phenomenon is referred to as *reward hacking* which we have illustrated empirically in the main text. Moreover, we show that distributional regularization (for simplicity, we focus on KL regularization) can effectively mitigate the danger of reward hacking.

**Theorem 1** (Reward Hacking). *Under the assumptions of Assumption 1, and let $\pi_{\hat{\theta}}$ be the policy learned to maximize $R(\tau)$*

$$\hat{\pi} = \arg\max_{\pi} \mathbb{E}_{\tau \sim \pi} R(\tau).$$

*Then for any $\epsilon > 0$, there exists an example such that the corresponding learned policy is at least $\epsilon$ worse than that of $\pi_{\mathrm{ref}}$:*

$$\mathbb{E}_{\tau \sim \hat{\pi}} R_*(\tau) \leq \mathbb{E}_{\tau \sim \pi_{\mathrm{ref}}} R_*(\tau) - \epsilon_{\mathrm{ood}} + \epsilon.$$

*Proof.* We consider the simple case that

$$\max_{\tau} R_*(\tau)$$

is achieved at $\tau_*$, and assume that for simplicity $\pi_{\mathrm{ref}}$ satisfies that

$$\mathbb{E}_{\tau \sim \pi_{\mathrm{ref}}} R_*(\tau) \geq R_*(\tau_*) - 0.5\epsilon.$$

Assume that for some $\hat{\tau}$, $R_*(\hat{\tau}) = R_*(\tau_*) - \epsilon_{\mathrm{ood}} + 0.5\epsilon$, and the reward model $R(\cdot)$ satisfies

$$R_*(\tau) \leq R(\tau) \leq \min\left(R_*(\tau_*) + 0.5\epsilon, R_*(\tau) + \epsilon_{\mathrm{ood}}\right),$$
$$R(\hat{\tau}) = R_*(\hat{\tau}) + \epsilon_{\mathrm{ood}}.$$

Then it is clear that $R(\hat{\tau}) = R_*(\tau_*) + 0.5\epsilon$, and thus the deterministic policy $\hat{\pi}$ that returns $\hat{\tau}$ achieves the optimal solution of

$$\max_{\pi} \mathbb{E}_{\tau \sim \pi} R(\tau).$$

It follows from the construction that

$$\mathbb{E}_{\tau \sim \hat{\pi}} R_*(\tau) = R_*(\hat{\pi}) = R_*(\tau_*) - \epsilon_{\mathrm{ood}} + 0.5\epsilon$$
$$\leq \mathbb{E}_{\tau \sim \pi_{\mathrm{ref}}} R_*(\tau) - \epsilon_{\mathrm{ood}} + \epsilon.$$

This proves the result. □

The previous result says that without regularization, reward hacking can lead to a degradation of the policy by $\epsilon_{\mathrm{ood}}$, and the degradation effect can be observed in practice (see the main text). On the other hand, with distributional regularization, this problem can be mitigated. We have the following result, which is stated for KL-regularization for simplicity. Note that results for otherwise $f$-divergence can also be obtained using a similar argument.

**Theorem 2** (KL-Regularization). *Consider the KL-regularized generation policy*

$$\hat{\pi}_{\eta} = \arg\max_{\pi} \left[ \mathbb{E}_{\tau \sim \pi} R(\tau) - \eta^{-1} D_{\mathrm{KL}}(\pi || \pi_{\mathrm{ref}}) \right],$$

*with $\eta \leq \epsilon_{\mathrm{ood}}^{-1}$. Let*

$$\pi_{*,\eta} = \arg\max_{\pi} \left[ \mathbb{E}_{\tau \sim \pi} R_*(\tau) - \eta^{-1} D_{\mathrm{KL}}(\pi || \pi_{\mathrm{ref}}) \right],$$

*and define*

$$\epsilon(\eta)^2 = \mathbb{E}_{\tau \sim \pi_{*,\eta}} (R(\tau) - R_*(\tau))^2.$$

*Assume that $R_*(\tau) \in [0, 1]$, then*

$$\mathbb{E}_{\tau \sim \hat{\pi}_{\eta}} R_*(\tau) \geq \mathbb{E}_{\tau \sim \pi_{*,\eta}} R_*(\tau) - 9\eta\epsilon(\eta).$$

*Proof.* For notation simplicity, we will drop subscript $\eta$ from $\hat{\pi}$ and $\pi_*$. Since it is well known that the closed form solution of $\hat{\pi}$ is

$$\hat{\pi}(\tau) = \frac{\exp\left(\eta R(\tau)\right)}{\mathbb{E}_{\tau \sim \pi_{\mathrm{ref}}} \exp\left(\eta R(\tau)\right)},$$

and the closed form solution of $\pi_*$ is

$$\pi_*(\tau) = \frac{\exp\left(\eta R_*(\tau)\right)}{\mathbb{E}_{\tau \sim \pi_{\mathrm{ref}}} \exp\left(\eta R_*(\tau)\right)},$$

we have

$$\hat{\pi}(\tau) = \frac{\exp\left(\eta \Delta R(\tau)\right)}{\mathbb{E}_{\tau \sim \pi_*} \exp\left(\eta \Delta R(\tau)\right)},$$

where $\Delta R(\tau) = R(\tau) - R_*(\tau)$.

Let $\mu = \mathbb{E}_{\tau \sim \pi_*} \exp\left(\eta \Delta R(\tau)\right)$, then we have

$$\mathbb{E}_{\tau \sim \pi_*} R_*(\tau) - \mathbb{E}_{\tau \sim \hat{\pi}} R_*(\tau)$$
$$= \mathbb{E}_{\tau \sim \pi_*} R_*(\tau) - \frac{\mathbb{E}_{\tau \sim \pi_*} \exp\left(\eta \Delta R(\tau)\right) R_*(\tau)}{\mathbb{E}_{\tau \sim \pi_*} \exp\left(\eta \Delta R(\tau)\right)}$$
$$\leq \frac{\mathbb{E}_{\tau \sim \pi_*} \left|\exp\left(\eta \Delta R(\tau)\right) - \mu\right| R_*(\tau)}{\mathbb{E}_{\tau \sim \pi_*} \exp\left(\eta \Delta R(\tau)\right)}$$
$$\leq \frac{\mathbb{E}_{\tau \sim \pi_*} \left|\exp\left(\eta \Delta R(\tau)\right) - \mu\right|}{\mathbb{E}_{\tau \sim \pi_*} \exp\left(-1\right)}$$
$$\leq e \cdot \mathbb{E}_{\tau \sim \pi_*}^{1/2} \left|\exp\left(\eta \Delta R(\tau)\right) - \mu\right|^2$$
$$\leq e \cdot \mathbb{E}_{\tau \sim \pi_*}^{1/2} \left|\exp\left(\eta \Delta R(\tau)\right) - 1\right|^2$$
$$\leq e^2 \eta \cdot \mathbb{E}_{\tau \sim \pi_*}^{1/2} \Delta R(\tau)^2$$
$$= e^2 \eta \epsilon(\eta).$$

This implies the bound. $\qquad \square$

We note that $\epsilon(0) = \epsilon_{\mathrm{id}}$, and $\epsilon(\eta) \leq \epsilon_{\mathrm{ood}}$ for all $\eta$. It follows that the suboptimality of regularized method in Theorem 2 is at worst

$$O(\eta \epsilon(\eta)) = O(\eta \epsilon_{\mathrm{ood}}),$$

and can be as good as

$$O(\eta \epsilon_{\mathrm{id}}).$$

Even in the worst case, when $\eta$ is small, this result significantly alleviates the worst case reward hacking of $\Theta(\epsilon_{\mathrm{ood}})$ in Theorem 1, demonstrating the effectiveness of distributional regularization in preventing reward hacking. This theory supports the empirical study we provided in the main text.

## A.2 Broader Impact Statement

Reinforcement Learning from Human Feedback (RLHF) offers a promising avenue for aligning consistency models with human preferences, enabling efficient and controllable generation. By optimizing models using human or proxy feedback, RLHF captures subjective qualities like aesthetics and relevance without relying on hand-crafted objectives. However, it also introduces risks—reward models may encode biases, and models can exploit them through reward hacking, especially given the fast convergence of consistency models. Despite these challenges, RLHF enhances sample efficiency and lowers inference costs compared to diffusion models. Realizing its full potential requires careful reward design, diverse feedback, and rigorous evaluation to ensure ethical and reliable deployment. Applying RLHF to consistency models enables faster and more controllable alignment with human preferences, offering practical gains in efficiency and sustainability. However, reliance on subjective or proxy rewards introduces risks of bias and reward hacking, which can be amplified by our approach. To ensure safe and effective deployment, this approach must be coupled with robust reward model design and diverse feedback sources.

### A.3 Additional Details

In this section, we outline the key training configurations for our models. All ROCM variants were trained with a batch size of 1 using 2 NVIDIA A6000 GPUs, and gradient accumulation was set to 1, resulting in an effective batch size of 1. The learning rate remained constant at $6 \times 10^{-5}$ throughout training. To ensure a fair comparison, baseline models were trained with the same learning rate. Additionally, we normalize the reward scores during training to bring them to order 0 for consistency across different reward models.

We report results in Table 1 after 10,000 training steps. For evaluation, we selected the best-performing checkpoints on their respective reward models that did not exhibit overfitting. Consistency model-based methods were trained and evaluated using 8 denoising steps, while diffusion-based methods used 20 steps. The optimal $\beta$ values used for each reward model during training are summarized in Table 2. We used publicly available base models from HuggingFace: `SimianLuo/LCM_Dreamshaper_v7` for consistency models and `Lykon/dreamshaper-7` for diffusion models.

Table 2: **Hyperparameters:** Optimal $\beta$ values for ROCM

| Regularization | $\beta$ | | | |
|---|---|---|---|---|
| | HPSv2 | PickScore | ImageReward | Aesthetic Score |
| JS-Divergence | 1000 | 1000 | 1000 | 2000 |
| KL-Divergence | $10^{-4}$ | $10^{-4}$ | $10^{-5}$ | $10^{-4}$ |
| Hellinger | $10^{-2}$ | $10^{-2}$ | $5 * 10^{-2}$ | $10^{-2}$ |
| Fisher Divergence | $5 * 10^{-5}$ | $5 * 10^{-5}$ | $10^{-4}$ | $5 * 10^{-5}$ |

We observed another intriguing effect of regularization, as illustrated in Fig: 8. Each regularization method guides the model toward a specific generation style, while the Aesthetic Score reward model consistently assigns high scores to all of them, demonstrating its generality. Interestingly, we also find that unregularized methods produce highly noisy outputs, whereas regularized methods generate relatively coherent images, each exhibiting its own distinct style even when overfitting.

### A.4 Algorithms

In Algorithm 1 we present the consistency multi-inference algorithm Luo et al. (2023) and in Algorithm 2 we present our training algorithm:

---
**Algorithm 1** Consistency Model K-step Generation

---
1: Draw $x_K = \epsilon_K \sim \mathcal{N}(0, I)$
2: $c \sim C$, where $C$ is the set of conditions (eg. prompts)
3: **for** $k = K, \ldots, 1$ **do**
4:      $\tilde{x}_k = f_\theta(x_k, \omega, c, t_k)$
5:      $\epsilon_{k-1} \sim \mathcal{N}(0, I)$
6:      $x_{k-1} = \alpha_{t_{k-1}} \tilde{x}_k + \beta_{t_{k-1}} \epsilon_{k-1}$
7: **end for**
8: **return** $x_0$

---

### A.5 $f$-divergence

In Table: 3 we present the table of $f$-divergences with their actual formula and closed form solution when the distributions are assumed to be Gaussian with different means and same standard deviation.

---

**Algorithm 2** Optimization with Divergence Regularization

---

1: Initialize parameters $\theta$
2: Set reference parameters $\theta_{\text{ref}} := \theta$
3: Set batch size $B$ and regularization weight $\lambda$
4: Set reward model $R(\cdot)$ and divergence $\mathcal{D}$
5: **repeat**
6:     Sample a batch of conditions $\{c_i\}$ of size $B$
7:     **for** $i = 1, \ldots, B$ **do**
8:         Sample noise $\epsilon^{(i)} \sim \mathcal{N}(0, I)$
9:         Generate trajectory $\{(x_k^{(i)}, \tilde{x}_k^{(i)})\}_{k=1}^{K} = G(\theta, \epsilon^{(i)}, c_i)$
10:        $R_i = R(\tilde{x}_1^{(i)}, c_i)$
11:        $\mathcal{D}_i = \sum_{k=2}^{K} \mathcal{D}_f(p_k(\cdot|\theta, x_k^{(i)}, c_i)||p_k(\cdot|\theta_{\text{ref}}, x_k^{(i)}, c_i))$
12:    **end for**
13:    Update parameters using the objective:

$$\theta \leftarrow \theta + \eta \nabla_\theta \left( \frac{1}{B} \sum_{i=1}^{B} \left[ R_i - \lambda \mathcal{D}_i \right] \right)$$

14: **until** Convergence

---

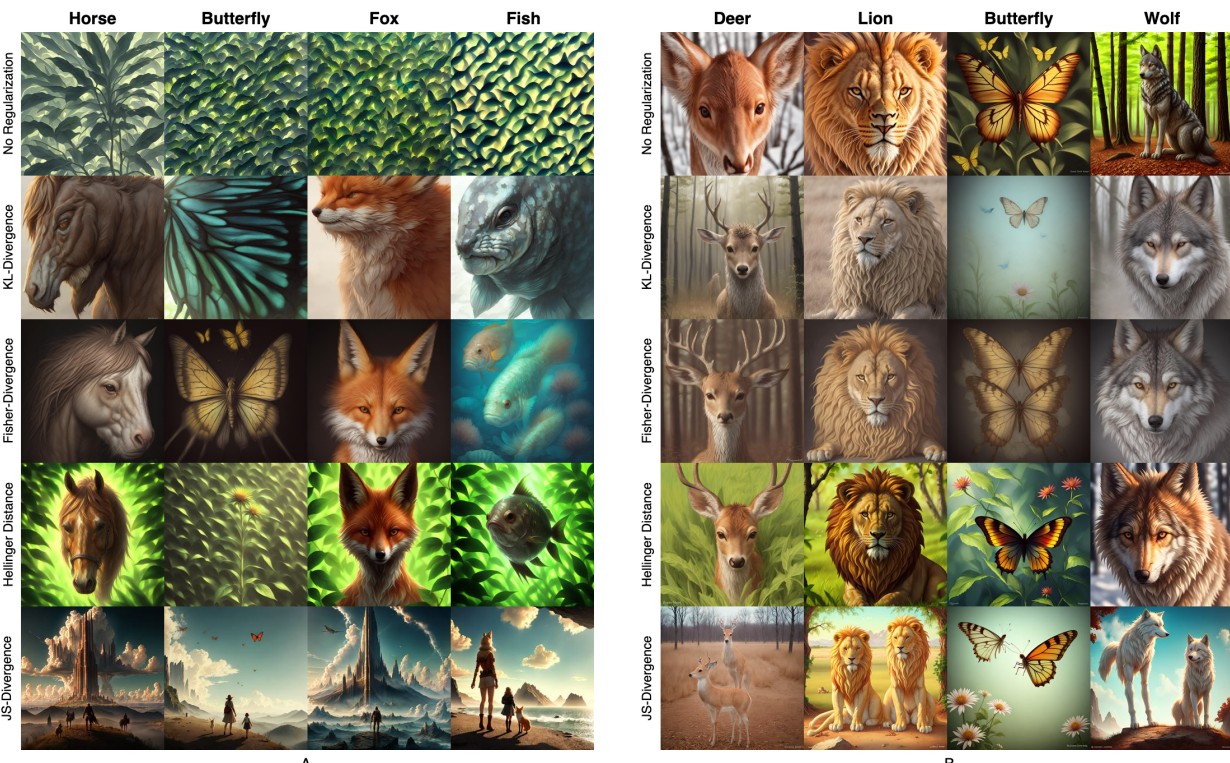

Figure 8: **Overfitting Analysis:** Figures A and B present results using the Aesthetic Score Wang et al. (2022) as the reward model. Figure B shows the best outputs from each method, illustrating how different divergences lead to distinct generation styles that are still favored by the reward model. Figure A highlights overfitted examples, revealing divergence-specific reward hacking behaviors. Regularization mitigates these effects—methods without it produce incoherent images, while regularized variants maintain legibility despite imperfect backgrounds.

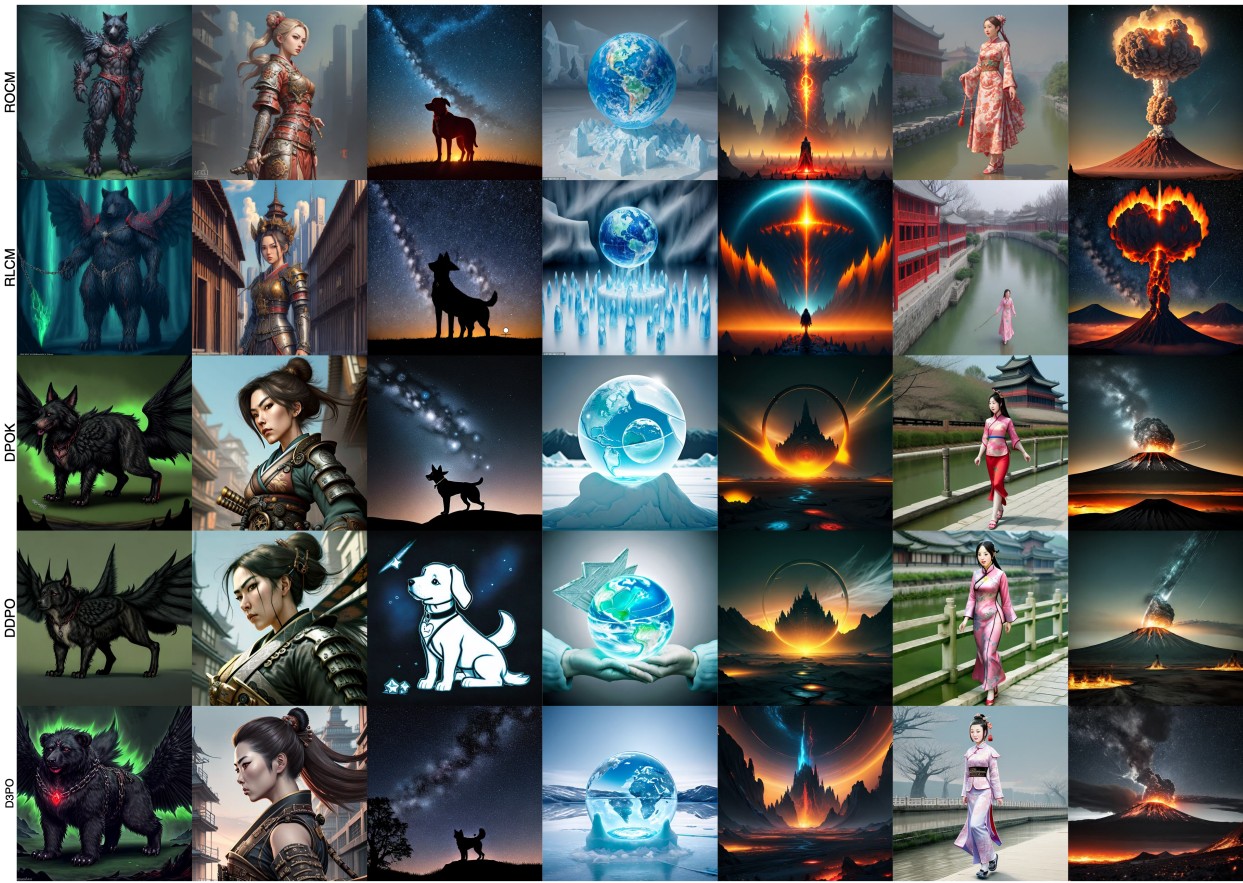

Figure 9: **Generation Comparison:** Sample Images generated by our baselines and ROCM trained on HPSv2 as reward model.

Table 3: $f$-**divergence:** This table summarizes the commonly used $f$-divergence. Here $x = \frac{p(t)}{q(t)}$. JS-Divergence doesn't have a closed-form solution for two Gaussian distributions.

| $f$-divergence | $f(x)$ | $D(\mathcal{N}(\mu_1, \sigma^2 I), \mathcal{N}(\mu_2, \sigma^2 I))$ |
|---|---|---|
| KL-Divergence | $x \log x$ | $\frac{\|\mu_1 - \mu_2\|_2^2}{2\sigma^2}$ |
| Hellinger | $(\sqrt{x} - 1)^2$ | $1 - \exp\frac{\|\mu_1 - \mu_2\|_2^2}{8\sigma^2}$ |
| JS-Divergence | $\frac{1}{2}(x \log \frac{2x}{x+1} + \log(\frac{2}{x+1}))$ | N/A |
| Fisher Divergence | $\|\log x\|^2 \, dx$ | $\frac{\|\mu_1 - \mu_2\|_2^2}{\sigma^4}$ |

Table 4: **Pretained Model Results:** Performance metrics of baseline models for diffusion and consistency based methods.

| Method | PickScore | Aesthetic | HPSv2 | ImageReward |
|---|---|---|---|---|
| Diffusion | 21.778 | 6.143 | 0.307 | 0.851 |
| LCM | 21.533 | 6.237 | 0.280 | 0.380 |

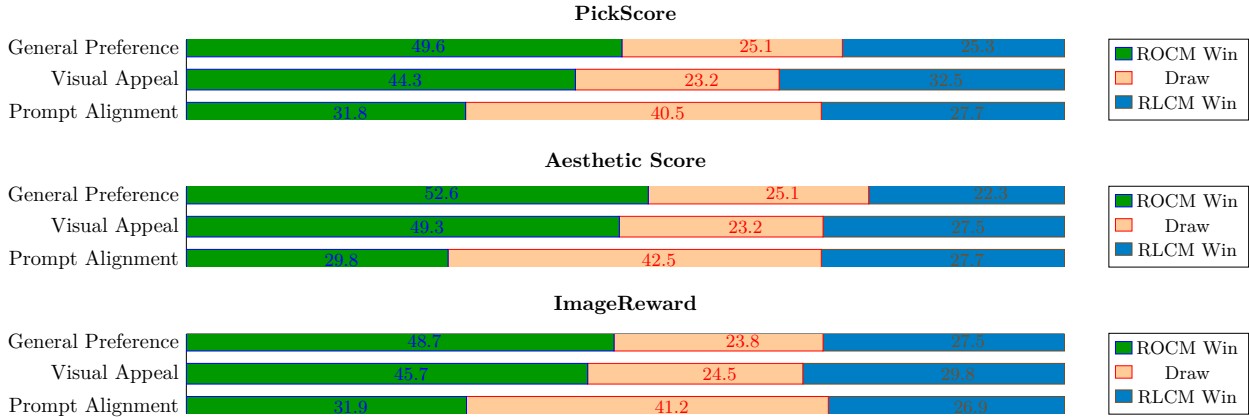

Figure 10: **Additional User Study Results:** Supplementing the HPSv2 study in the main paper, we present user survey outcomes for PickScore, Aesthetic Score, and ImageReward.

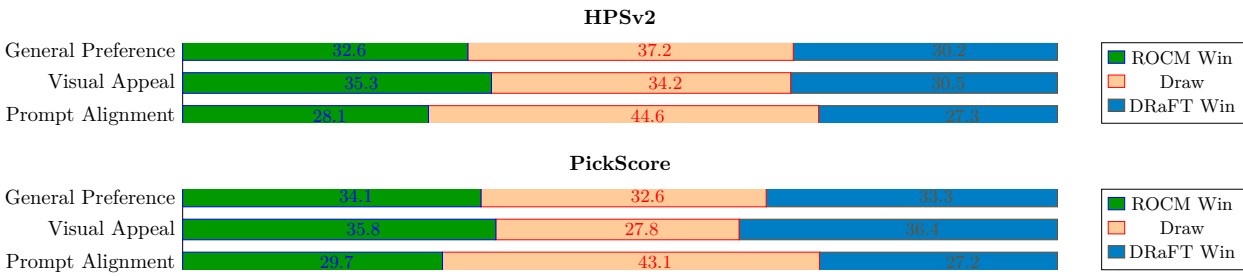

Figure 11: **Additional User Study Results:** Human study results on HPSv2 and PickScore for DRaFT and ROCM.

