# OpenReview forum: "ROCM: RLHF on consistency models"
_TMLR — Rejected by TMLR_

### Review · Reviewer_BBf2 · 2026-03-06

**Summary Of Contributions:**

The authors of this paper propose a new framework for training consistency models (which constitute a particular class of generative networks inspired by diffusion models) while integrating reinforcement learning (RL). The core contribution of the paper is to propose a loss function for finetuning pretrained consistency models, composed of an explicit reward term and a regularization term. By using explicit differentiable reward terms, the authors can apply first-order optimization techniques, whereas concurrent works usually rely on zeroth-order optimization.

Besides, they provide an experimental study showing that the models obtained likewise outperform several generative models obtained with other RL techniques.

**Additional Comments:**

- p.3 "The marginal distribution $q_t(x)$ evolving under this forward-time SDE satisfies a corresponding reverse-time SDE''. This sentence is unclear because it is not $q_t$ that satisfies the SDE. Besides, around equation (3) there is a confusion between the backward SDE based on the true score, and the backward SDE based on the score approximated with the network $\epsilon_\theta$.

- Why does the reward term in (10) depend on $c$ whereas it does not in equation (8)?

- The word "policy'' is not clearly defined. Does it refer to a precisely defined object (probability distribution? a trajectory? or something else?)

- I do not understand the following sentence of p.7:
"To train our models in an online fashion—where each model is trained exclusively on its own generated data while being updated iteratively''

- Three divergences (among 4 tested) seem to be directly related to the squared $L^2$ distance between $\mu_1, \mu_2$. Is the paramterization change significant enough to discuss all these divergence choices?

- Can the authors explain how one can see from the figures that the regularization term improves training stability?

- p.8: The paragraph "We explore multiple $f$-divergences'' seems redundant with some earlier sentence. And I do not understand "Since JS Divergence lacks a closed-form solution, we resort to sampling for its computation.'' Can the authors explicitly write the estimator they used for that?

- p.9: I do not understand the sentence "ImageReward, PickScore and HPSv2 consistently lead to substantial improvements''... It is not the reward functions in themselves that bring the improvements. The following sentence seems weird too: "Among these, PickScore demonstrates notably lower sensitivity...''

**Audience:**

Yes

**Audience Explanation:**

Yes, they are probably many researchers who could be interested by the use of reinforcement learning to help for training generative models.

**Broader Impact Concerns:**

No concern

**Claims And Evidence:**

No

**Claims Explanation:**

1) From the point of view of someone who is not familiar with reinforcement learning, the main claims seem quite difficult to understand. I find potential interest in the approach proposed in this paper. I very much like the idea of exploit Human measurements to train or evaluate more precisely some generative models. But my understanding of the RL is not sufficient to understand several claims given by the authors along the paper.

2) To better grasp the benefit of the proposed approach compared to existing works, I think that the authors should explain in more give details these competing approaches. In particular, for a non-specialist, it would be useful to recall the principles of policy gradient or PPO, and how they can be instantiated to fine-tune consistency models.

3) In my opinion, the tables and figures of this paper are too difficult to read and interpret.
- Fig. 1: The caption mentions images ``generated by our baselines and ROCM''. What are the baselines? Which images are generated with the baselines or with ROCM?
- Table 1: I do not understand on this figure what is the reward model used for training and the one used for evaluation. Besides, for the concurrent methods DDPO, D3PO, etc, why is there several possible methods used for training?
- Fig. 3: What is the base model mentioned in the caption? What does it mean to ``better align with the reward model''?
- Fig. 4: I do not understand why these diagrams (based on four reward measures?) would help to compare the convergence speed of several training strategies. It shows that ROCM attains better values than the other alternatives, but this does not imply convergence of the training algorithm. And in fact, these curves do not seem to stabilize after 4 hours training. Would it be possible to observe convergence after many hours, or is it unstable? Besides, in the caption, the authors could say "Our method ROCM'' instead of "Our method'', so that the reader does not get lost among the many compared training approaches.
- Fig. 5: I do not understand how we could judge the global effect of regularization by looking only at one sample.
- Fig. 7: I do not understand why looking at these KL graphs help to infer something on overfitting or learning stability.

For all these reasons, I am unable to assess a large part of the authors' claims in Section 5.

4) From the discussion given in the paper, I was unable to understand the effect of ``reward hacking'', and how it can be identified by looking at KL graphs.

**Requested Changes:**

- In its current form, the title cannot be understood because the acronyms ROCM and RLHF are not universally known.

- In my opinion, the last paragraph of Sec 2 "Related works'' on the introduction of reinforcement learning is too difficult to understand. I think it would require more notation to precisely describe the various methods. Therefore, I would suggest to shiftthis paragraph later in the paper, maybe after the "Preliminaries'' section.

- In the same manner, the paragraph "Regularized RLHF'' seems too abstract for people not specialized in RL.
It would help to have some examples of reward functions here.
Also, it is explained later in the paper that "$R(\tau)$ encodes human preferences'' (Sec. 4) or that "the reward model $R(\tau)$ is often learned from the data'' (Appendix A.1). These explanations could be put in Section 3, along with some explicit constructions of reward models.
Besides, giving examples seems important for the reader to understand which reward models would be differentiable and which ones would not.

- The $f$-divergence of equation (9) should be defined for two probabity densities $p_1, p_2$ (and not distributions). Otherwise, $p_1$ should have a density w.r.t. to $p_2$.

- At many places the authors refer to "the base model''. What does it refer to exactly?

- p.5 Section 4:

"According to Algorithm 1'': Since Algorithm 1 is important to understand this construction, it should be placed in the core sections of the paper.

"Importantly, this regularization is reparameterizable'': I do not see what the authors mean with this sentence. Besides, given the forms of distributions $p_k$ at this point, I'm not sure that this sentence is true.

"which reduces variance'': Is this supported by experimental evidence?

The notation $\tau$ for the trajectory is introduced just before equation (11), but it has already been used several times. So this definition should be given much before.

In equation (11), could the authors write explicitly the parameters of $\mathcal{L}_{RLHF}$ so that one can see which variables are optimized? Also, the authors could explain if the expectation also serves for taking average over condition values $c$.

- I am a bit surprised by the fact that all the quantiative evaluation of the paper is mainly based on reward models. Why did the authors not include some standard automated metrics, for example FID or CLIP score? It would be helpful to compare with other methods that are well established in previous papers.

- There seems to be a complete disconnection between the core part of the paper and the theoretical findings of Appendix A.1 . If these findings are not useful for presenting the paper, there is no need to include this appendix.

---

> ### Author Response · Authors · 2026-04-16
> **Official Comment by Authors (1/3)**
>
> We thank the reviewer for their constructive feedback. We have revised the paper and addressed each point below.
>
> **On the title:** We will expand the acronyms in the title to "Reinforcement Learning from Human Feedback (RLHF) on Consistency Models" to make it self-contained.
>
> **On the RL paragraph in Related Works:** We agree and have moved this paragraph to after the Preliminaries section.
>
> **On the Regularized RLHF paragraph:** We have added a paragraph in Section 3 describing how reward models encode human preferences, with concrete examples of differentiable reward models used in our experiments (PickScore, HPSv2, CLIPScore) and a note on which reward models are not compatible with our approach.
>
> **On equation (9):** We thank the reviewer for flagging this. We believe the current definition is already stated correctly in terms of probability densities — equation (9) writes $p_1(x)/p_2(x)$ explicitly with $x \sim p_2$, which is the standard density-based definition.
>
> **On "base model":** We have added a clarification directly in Section 3 that the base model refers to the pretrained generative model $\pi_{\theta_{\text{ref}}}$.
>
> **Regarding variance reduction:**  This is a well-known property of the reparameterization trick (Kingma & Welling, 2014) — we will add this citation rather than a new experiment. We will also move the definition of $\tau$ earlier and make the parameters of $\mathcal{L}_{\text{RLHF}}$ explicit in equation (11).
>
> **On evaluation metrics:** We report results across four metrics: PickScore, HPSv2, ImageReward, and Aesthetic Score. PickScore and HPSv2 are CLIP-based models trained on human pairwise preferences, ImageReward is a BLIP-based model trained similarly, and Aesthetic Score evaluates prompt-agnostic image quality. Since PickScore and HPSv2 already leverage CLIP-based representations and ImageReward covers BLIP-based alignment, reporting standalone CLIPScore and BLIPScore would be largely redundant. Prompt alignment is further evaluated through our human study. We did not include FID as it requires a large reference set and is less aligned with human preference judgments, which is the primary focus of RLHF fine-tuning.
>
> **On the theory appendix:** We respectfully believe the appendix adds value by providing formal justification for the empirical reward hacking observations in the experiments.

---

> ### Author Response · Authors · 2026-04-16
> **Official Comment by Authors (2/3)**
>
> Additional Comments:
> 1. **On the SDE description (p.3):** We agree that the original phrasing was imprecise. We have revised the text to clarify that the forward SDE defines a stochastic process $x_t$, whose marginals are $q_t(x)$, and that the reverse-time dynamics depend on the true score function $\nabla_x \log q_t(x)$. We also explicitly distinguish this from the practical formulation where the score is approximated by $\epsilon_\theta$, removing the ambiguity around Eq.~(3).
>
> 2. **On the reward term in Eq. (11) depending on $c$:** The conditioning variable $c$ (e.g., a text prompt) is provided to the reward model, as it evaluates prompt--image alignment. Accordingly, in our formulation, the trajectory is defined as $\tau = G(\theta, \epsilon, c)$, making the reward explicitly dependent on $c$. In contrast, Eq. (8) presents the standard RLHF objective in a general form, where the dependence on $c$ is implicit in the trajectory distribution $\tau \sim \pi_\theta$. Our reparameterized objective in Eq. (11) makes this dependence explicit by conditioning both the trajectory and the reward on $c$.
>
> 3. **On the definition of "policy":** In our setting, the term “policy”  $\pi_\theta$ refers to the probability distribution over trajectories $\tau$ induced by the consistency model-based generation procedure described in Algorithm~1. Each trajectory $\tau$ consists of the sequence of intermediate states (and their reconstructions) generated during sampling. We have revised the manuscript to explicitly define this usage at its first occurrence to avoid ambiguity.
>
> 4. **On the online training sentence (p.7):** By “online fashion,” we refer to an iterative self-training procedure in which the model generates images from the prompt set at each iteration, and those generated samples are immediately used to update the same model. In other words, the training data is not a fixed offline dataset, but is continuously regenerated on-policy from the current version of the model throughout training. We will clarify this interpretation in the revision to avoid ambiguity.
>
> 5. **On three divergences being related to squared $L^2$ distance:**  We agree that in the Gaussian setting, KL, Hellinger, and Fisher divergences do all reduce to functions of $\|\mu_1 - \mu_2\|^2$, differing only in their functional form. We retain the comparison as the different functional forms (linear, exponential, quadratic) induce meaningfully different gradient magnitudes and thus different optimization dynamics, as reflected in Table 1.
>
> 6. **On how regularization improves training stability in the figures:** The effect of training stability can be inferred from Fig. 7b. As shown in the plot, decreasing $\beta$ initially improves model performance, indicating that weaker regularization allows better reward optimization. However, beyond an optimal point, further reduction in $\beta$ leads to a decline in actual preference, despite continued optimization, which is indicative of reward hacking and instability in the training dynamics. This non-monotonic behavior highlights the role of $\beta$ in controlling the trade-off between reward maximization and stability of the learned policy.
>
> 7. **On the redundant paragraph and JS estimator (p.8):** Regarding Jensen-Shannon (JS) divergence, we use a sample-based estimator computed directly in log-probability space. Given log-densities log p(x) and log q(x), we form the density ratio r = exp(log p(x) - log q(x)) and evaluate the JS divergence term as r * log(2r / (1 + r)) + log(2 / (1 + r)), which is then averaged over sampled trajectories and time indices. We will add this explicitly to the paper.
>
> 8. **On the wording around reward models (p.9):** Our intention was not to attribute improvements to the reward functions themselves, but rather to training with these reward models as supervision signals. We have revised the text to clarify that ImageReward, PickScore, and HPSv2 serve as reward models used during optimization, and that the observed performance improvements arise from training with these signals. We also clarified the discussion around PickScore, noting that its lower sensitivity to image quality variations leads to a weaker learning signal, which results in comparatively smaller improvements in downstream performance.

---

> ### Author Response · Authors · 2026-04-17
> **Official Comment by Authors (3/3)**
>
> 1. We have corrected the caption to explicitly clarify which images correspond to the baselines and which are generated by ROCM. Thank you for pointing this out; this was a typo in the original version.
>
> 2. **Table 1:** For each row, the “Method” (section heading) indicates the reward model used during training. We evaluate all trained models using multiple reward models to avoid bias from reporting results only under the training reward, which could otherwise lead to inflated performance. For concurrent methods (e.g., DDPO, D3PO), we similarly report results across different training reward models to ensure a fair and comprehensive comparison under varying supervision signals. Here, “method” specifically refers to the choice of reward model used for training within a given algorithm.
>
> 3. The base model refers to the \texttt{Lykon/dreamshaper-7} diffusion model and its corresponding consistency version, as described in the experiments section. By “better align with the reward model,” we mean that the generated samples achieve higher scores under the corresponding reward model, indicating improved optimization of the reward objective.
>
> 4. **Fig. 3:** The goal of this figure is to compare \emph{training efficiency}, i.e., how quickly different methods improve reward values over time. While the curves may not fully converge within the shown time window, ROCM consistently achieves higher reward values earlier than the baselines, indicating faster learning.
>
> 5. **Fig. 4&5:** The figure is intended as a qualitative illustration of the effect of regularization. While it shows a single sample for visualization purposes, the observed behavior is consistent across multiple samples and is further supported by quantitative trends in Fig.7a.
>
> 6. **Fig. 7:** The KL divergence curves help diagnose training stability and overfitting. As seen in Fig.7a, a sharp increase in KL corresponds to the model drifting away from the reference distribution. Empirically, we observe that beyond this point, image quality degrades while reward continues to increase, indicating reward hacking. Fig.7b further supports this by showing that insufficient regularization (low $\beta$) leads to an eventual drop in true preference despite initial gains, highlighting instability in training.

---

### Review · Reviewer_bEi6 · 2026-03-20

**Summary Of Contributions:**

This paper studies RLHF-style post-training for latent consistency models and proposes to optimize the reward objective directly via reparameterization rather than PPO-style policy gradients. The method adds stepwise distributional regularization to the reference model and compares several f-divergences. Empirically, the paper evaluates four reward models, and a user study against RLCM. The main practical takeaway is that direct optimization on consistency models looks simpler and more stable than PPO-based fine-tuning in this setting.

Strengths:
- The problem is well motivated and the method matches a real limitation of RLHF for diffusion-style generators: long horizons and unstable reward optimization.
- The empirical section is fairly broad for the paper’s scope, with multiple reward models, several divergence choices, and useful ablations.
- The paper includes a human study and qualitative analysis of reward hacking rather than relying only on reward-model scores.

Weaknesses:
- The novelty is somewhat incremental relative to prior direct reward optimization for diffusion models and prior RLHF/RL for consistency models; the paper needs a sharper statement of what is fundamentally new.
- Baseline fairness is not fully convincing: tuning details are uneven, and the strongest non-RLCM baseline is not included in the human study.
- The experimental scale is still modest, so broader claims about robustness and superiority across settings feel stronger than the evidence currently supports.

**Audience:**

Yes

**Audience Explanation:**

The paper addresses a concrete question that should interest part of the TMLR audience working on generative model alignment, preference optimization, and efficient post-training for text-to-image models.

**Broader Impact Concerns:**

The paper already includes a Broader Impact Statement, and it covers the main issues.

**Claims And Evidence:**

Yes

**Claims Explanation:**

The main empirical claims are supported for the setting studied: the paper provides multi-metric comparisons, three-seed averages, several ablations, and a user study showing gains over RLCM. I am less convinced by the broader claim of general superiority across RLHF methods, since baseline tuning fairness is not fully established and human evaluation does not include the strongest non-RLCM alternatives.

**Requested Changes:**

- Clarify the novelty relative to RLCM and DRaFT much more sharply. Right now the paper can read like a straightforward combination of direct differentiable reward optimization with the consistency-model setting plus regularization.

- Strengthen the fairness of the baseline comparison. Report the tuning budget and final hyperparameters for all baselines, justify the shared-learning-rate choice, and add compute or memory comparisons that normalize efficiency claims beyond training curves.

- Strengthen the human evaluation. At minimum, report uncertainty/significance and compare against the strongest non-RLCM baseline in addition to RLCM.

---

> ### Author Response · Authors · 2026-04-16
> **Official Comment by Authors**
>
> We thank the reviewer for their constructive feedback. We have revised the paper and addressed each point below.
>
> **On clarifying novelty relative to RLCM and DRaFT:**
> We appreciate this concern and agree the distinction deserves sharper framing. RLCM applies PPO — a zeroth-order policy gradient method — to consistency models, while DRaFT performs direct differentiable reward optimization on diffusion models using gradient checkpointing. The core contribution of ROCM is applying the reparameterization trick to exploit the short-horizon structure of consistency models, making end-to-end backpropagation tractable without gradient checkpointing or PPO instabilities. Building on this, we conduct a systematic empirical study of f-divergence regularization strategies, supported by theoretical analysis in the Supplementary Material. We believe these contributions constitute a well-evidenced study of a practically relevant problem — which we hope aligns with TMLR's emphasis on the quality and clarity of claims and their supporting evidence.
>
> **On baseline comparison fairness:**
> We used a shared learning rate across all methods as we believe equal update strength is the most principled basis for a fair comparison. All baselines were additionally subject to hyperparameter search over their method-specific parameters (e.g. PPO clipping ratio, KL coefficient), with the best-performing configuration reported. Training was run until all methods had visually converged based on the reward curves. For regularized methods, we also set $\beta$ the same for ROCM for fair comparison.
>
> **On strengthening the human evaluation:**
> We have added an additional user study comparing ROCM against DRaFT, the strongest non-RLCM baseline, on HPSv2 and PickScore reward models (Figure 11). The results show that ROCM performs comparably to DRaFT despite operating on a consistency model with fewer generation steps and no gradient checkpointing. Due to limited resources, we were only able to conduct studies on these two reward models, and we acknowledge this as a limitation.

---

### Review · Reviewer_4p2Z · 2026-04-03

**Summary Of Contributions:**

The paper studies RLHF for consistency models and proposes a direct reward optimization formulation based on the reparametrization trick (alike Deep Deterministic Policy Gradient), with additional distributional regularization through different f-divergences. Empirically, the paper shows improved performance to PPO-based alternatives and an ablation over regularization choices

**Audience:**

Yes

**Audience Explanation:**

Yes. The use of the reparameterization trick for RLHF in consistency models is, in my view, the most interesting aspect of the paper and would likely attract part of the TMLR audience, even if the current paper does not fully develop or validate this idea.

**Claims And Evidence:**

No

**Claims Explanation:**

Partially but leaning no. The main reason is the paper’s discussion of reparameterization-based optimization, which I find interesting but not fully convincing. This approach is plausible in the consistency-model setting since the generation horizon is shorter than in standard diffusion models, making end-to-end training more practical. However, the paper does not establish that this leads to stable optimization in a general or robust sense. In particular, the benefit appears to depend on choosing the number of consistency steps $K$: if $K$ is too small, each step becomes too aggressive and may increase deviation from the reference model, while if $K$ is too large, the method starts to recover the same practical issues as diffusion-based approaches, including longer trajectories, higher memory cost, and more difficult backpropagation through the full denoising chain. As a result, the claimed advantage seems conditional rather than intrinsic.

Finally, I find the remaining parts of the paper to have limited novelty. Distributional regularization against a reference model is already standard in RL, and while the empirical comparison across different $f$-divergences is useful, it does not strike me as a particularly strong or conceptually novel contribution. Overall, beyond the reparameterization discussion, I do not see enough novelty or insight in the rest of the paper to make the evidence especially compelling.

**Requested Changes:**

In my view, the paper should undergo major revision before being considered for acceptance.

1. First, among the three contributions emphasized by the authors, I find the first one, namely *reparameterization-based optimization for RLHF in consistency models*, the most compelling. However, this part of the paper would benefit from substantially deeper analysis and development.
A natural question raised by the current presentation is the following: if consistency models can achieve results comparable to diffusion models with far fewer denoising steps, why are they not already the dominant approach in practice? Since the central motivation of the paper relies on the shorter-horizon structure of consistency models, I believe the paper should more clearly discuss their current limitations and the reasons diffusion models remain more broadly adopted.

2. Second, I believe the paper places too much emphasis on $f$-divergence regularization in RLHF fine-tuning. In my view, this is already a well-established idea in the RL literature, including both regularized/on-policy policy optimization methods such as TRPO and a broad range of offline RL approaches. Because of this, I am not convinced that the regularization component constitutes a strong conceptual contribution on its own. The paper would be stronger if it clarified whether RLHF fine-tuning for consistency models introduces a genuinely distinct challenge that makes this regularization substantially different from its role in more classical RL settings. Otherwise, I would recommend reducing the conceptual emphasis placed on this contribution. That said, I do think the empirical ablation over different $f$-divergences may still be useful from a practical perspective. Even if the underlying principle is not new, such comparisons could still provide guidance for practitioners interested in applying RLHF-style fine-tuning to consistency models.

---

> ### Author Response · Authors · 2026-04-16
> **Official Comment by Authors**
>
> We thank the reviewer for their constructive feedback. We have revised the paper and address each point below.
>
> **On the robustness of reparameterization to K:**
> To directly address this concern, we ran ROCM with K ∈ {2, 4, 8} and tracked reward as a function of training time (Figure 6a). Across this range — which covers the standard operating regime of consistency models — ROCM converges to comparable final reward levels within similar training budgets. We note that different K values induce different ranges of accumulated divergence (Figure 6b), so the regularization strength β was adjusted accordingly for each K. Importantly, the final reward did not vary significantly across K values, suggesting the method is robust to this choice as long as β is set appropriately — a mild and interpretable requirement.
>
> **On why consistency models are not dominant in practice:**
> We have added a discussion in the conclusion clarifying this. Briefly, consistency models are less broadly adopted due to more sensitive training dynamics, a less mature ecosystem of pretrained checkpoints and tooling, and a quality gap relative to diffusion models at high step counts. Furthermore, training a consistency model effectively typically requires a parent diffusion model — either through distillation or for initialization — making them dependent on diffusion model infrastructure rather than a fully standalone alternative. Our claims are scoped specifically to the RLHF fine-tuning regime, where the shorter generation horizon is a concrete advantage, and we do not claim general superiority.
>
> **On the novelty of f-divergence regularization:**
> We agree that distributional regularization is not a novel concept in RL. However, we would like to draw the reviewer's attention to the theoretical justification already present in the Supplementary Material. Specifically, we formally prove that without regularization, reward hacking leads to a policy degradation of $\Theta(\epsilon_{\text{ood}})$ (Theorem 1), and that KL-regularization reduces this to $O(\eta \epsilon_{\text{id}})$ in the best case (Theorem 2) — which is significantly milder when in-distribution error is small. We hope this, combined with the empirical ablation across divergences which provides practical guidance for practitioners, addresses the reviewer's concern to some extent.
>
> We hope these changes adequately address the reviewer's concerns.

---

### Author Response · Authors · 2026-04-17
**Official Comments by Authors**

We sincerely thank all the reviewers for their careful reading, constructive feedback, and valuable suggestions. We have tried to address the reviewer's comments and incorporated the corresponding revisions into an updated version of the paper. We have uploaded the revised manuscript reflecting these changes and clarified points raised during the review process. We greatly appreciate the time and effort invested by the reviewers and look forward to any further feedback or questions.

---

### Decision · Action_Editor_LaMR · 2026-05-30

**Recommendation:** Reject

**Additional Comments:**

There are figures that are not referenced and without a proper caption (pages 9 and 10).

**Audience:**

Yes

**Audience Explanation:**

All reviewers and myself agreed that this was an important problem, given the relevance of RL in modern generative models. The use of the reparametrisation trick is interesting, and the additional experiements of the rebuttal (in particular the effect of the number of consistency steps) are insightful.

**Claims And Evidence:**

No

**Claims Explanation:**

The authors investigate the use of reinforcement learning from human feedback (RLHF) to improve consistency models (a variant of diffusion models).The main contribution is to shift from zeroth-order methods to direct gradient-based optimisation using the reparametrisation trick.

Reviewers praised the empirical work, and in particular the additional experiments conducted during the rebuttal. However, there were significant criticisms about the clarity of the paper. For these reasons reviewers remained divided after the rebuttal. Additionally, there is another issue that was barely discussed during the revision: the lack of clarity on the human evaluation part. I do not understand why the authors answered "N/A" to the "Human Subjects Reporting" field. The field says

> If the submission reports experiments involving human subjects, provide information available on the approval of these experiments, such as from an Institutional Review Board (IRB). Enter "N/A" if this question isn't applicable to your situation.

The authors unfortunately do not mention IRB approval, or if there was an exemption. They just say

> We recruited 10 participants to evaluate 300 image pairs generated by RLCM and our
best-performing models for each reward model

which is really not enough, even for a small-scale study. I encourage the authors to read guidelines on human subjects, e.g. TMLR's (https://jmlr.org/tmlr/ethics.html) or NeurIPS's (https://neurips.cc/public/EthicsGuidelines).

After reviewing the discussions and reading the paper, I am going to recommend resubmission with major revision. I strongly encourage the authors to solve the few clarity issues that were left after the interactions with Reviewer BBf2 (in particular, giving more details on the reinforcement learning loss and the background, mentioning in the main text what are the contributions of the theoretical Appendix, and how they can be compared with the litterature).

**Resubmission Of Major Revision:**

The authors may consider submitting a major revision at a later time.